# Characteristics of BRCA2 Mutated Prostate Cancer at Presentation

**DOI:** 10.3390/ijms232113426

**Published:** 2022-11-03

**Authors:** Hyunho Han, Cheol Keun Park, Nam Hoon Cho, Jongsoo Lee, Won Sik Jang, Won Sik Ham, Young Deuk Choi, Kang Su Cho

**Affiliations:** 1Department of Urology, Urological Science Institute, Yonsei University College of Medicine, Seoul 03722, Korea; 2Pathology Center, Seegene Medical Foundation, Seoul 04805, Korea; 3Department of Pathology, Yonsei University College of Medicine, Seoul 03722, Korea; 4Department of Urology, Prostate Cancer Center, Gangnam Severance Hospital, Yonsei University College of Medicine, Seoul 06229, Korea

**Keywords:** prostate cancer, *BRCA2*, PSA

## Abstract

Genetic alterations of DNA repair genes, particularly BRCA2 in patients with prostate cancer, are associated with aggressive behavior of the disease. It has reached consensus that somatic and germline tests are necessary when treating advanced prostate cancer patients. Yet, it is unclear whether the mutations are associated with any presenting clinical features. We assessed the incidences and characteristics of *BRCA2* mutated cancers by targeted sequencing in 126 sets of advanced prostate cancer tissue sequencing data. At the time of diagnosis, cT3/4, N1 and M1 stages were 107 (85%), 54 (43%) and 35 (28%) samples, respectively. *BRCA2* alterations of clinical significance by AMP/ASCO/CAP criteria were found in 19 of 126 samples (15.1%). The *BRCA2* mutated cancer did not differ in the distributions of TNM stage, Gleason grade group or histological subtype compared to *BRCA2* wild-type cancers. Yet, they had higher tumor mutation burden, and higher frequency of *ATM* and *BRCA1* mutations (44% vs. 10%, *p* = 0.002 and 21% vs. 4%, *p* = 0.018, respectively). Of the metastatic subgroup (M1, n = 34), mean PSA was significantly lower in *BRCA2* mutated cancers than wild-type (*p* = 0.018). In the non-metastatic subgroup (M0, n = 64), PSA was not significantly different (*p* = 0.425). A similar trend was noted in multiple metastatic prostate cancer public datasets. We conclude that *BRCA2* mutated metastatic prostate cancers may present in an advanced stage with relatively low PSA.

## 1. Introduction

Germline mutations of DNA homologous recombination repair (HRR) genes, particularly *BRCA2*, are frequently found in advanced prostate cancers (PCs) [1,2], and predict poor outcomes for conventional therapies [3,4,5,6]. Therefore, PCa screening using serum prostate-specific antigen (PSA) is recommended in *BRCA1/2*-carrying males [7]. Also, somatic alterations of *BRCA1/2* are found in 5–10% of de novo metastatic cancers and 10–15% of metastatic castration-resistant prostate cancers (mCRPCs) [8,9]. Furthermore, *BRCA2* mutation in mCRPC is associated with poor response to docetaxel chemotherapies and new hormonal agents [5,10,11]. In contrast, recently, polyadenosine diphosphate-ribose polymerase (PARP) inhibitors olaparib and rucaparib have demonstrated improved overall survival in BRCA2-altered metastatic PCs, combined with or followed by conventional chemo-hormonal therapies [12,13,14]. This suggests that BRCA2-altered PCs are genetically and biologically distinguished from the rest, which is linked to drug sensitivity.

It has been known that some PCs present with low serum PSA but high-grade histology, which often rapidly progresses to lethal disease [15,16]. Reportedly, the PSA-low tumors are characterized by aberrant histology such as neuroendocrine, ductal/intraductal, or sarcomatoid features [15,17,18]. In particular, intraductal and cribriform histology are frequently found in *BRCA2* and other HRR gene-mutated PCs [19,20,21].

We hypothesized that *BRCA2* mutation—either somatic or germline—is a special subgroup of PC and may be clinically or pathologically distinguished from those wild-type tumors. In this paper, we reviewed a consecutive series of targeted gene panel sequencing data of high-risk PCs—Gleason grade group (GGG) 4–5, cT3/4 stage, N1 or M1 stage at diagnosis. We identified those with *BRCA2* mutations, and compared their clinical, pathological and genomics characteristics with the wild-type tumors.

## 2. Results

We evaluated 126 sets of consecutive prostate cancer tissue NGS data (from June 2019 and April 2021). There were 26 patients with 44 *BRCA2* genetic alterations (42 mutations, 1 amplification and 1 heterozygous loss) (Appendix A). Among the 26 patients, we identified 19 (15.1%) patients with *BRCA2* alterations that were classified as “Pathogenic” or “Likely Pathogenic” by ACMG classification and “Tier I” or “Tier II” by AMP/ASCO/CAP classification (Appendix A). Of note, the 1 hetloss was considered as an alteration whereas the 1 amplification was considered as wild-type, since *BRCA2* is a tumor-suppressor gene.

### 2.1. Clinical Characteristics of BRCA2 Mutated Prostate Cancers

We compared the clinical characteristics of the 19 cases of *BRCA2* oncogenic alterations with the remaining 107 wild-type cases. Patient age or TNM stage distribution were not significantly different between *BRCA2* mutated (*BRCA2*mut) and wild-type (*BRCA2*_WT_) tumors (Table 1). Interestingly, mean PSA at diagnosis of *BRCA2*mut group tended to be lower than that of the *BRCA2*_WT_ group (94.7 ng/mL vs. 242.7 ng/mL, *p* = 0.101, Table 1 and Appendix A).

### 2.2. Pathologic Characteristics of BRCA2 Mutated Prostate Cancers

Gleason grade group (GGG) 5 tumors were more frequent in *BRCA2*mut group than the *BRCA2*_WT_ group, yet not statistically significant (68% vs. 50%, *p* = 0.140, Table 1). Neuroendocrine/small histology was more frequent in *BRCA2*mut group, yet not statistically significant either (11% vs. 1%, *p* = 0.06, Table 1). Lastly, ductal adenocarcinoma was less significant in *BRCA2*mut group (5% vs. 15%, *p* = 0.06, Table 1).

### 2.3. PSA at Presentation by Initial Stage and by BRCA2 Mutation

We separated the cohort into metastasis at presentation (M1, n = 34) and no metastasis at presentation (M0, n = 92) subgroups. The mean serum PSAs at presentation were: 102 ng/mL (*BRCA2*mut M1, n = 8), 869 ng/mL (*BRCA2*_WT_ M1, n = 26), 89 ng/mL (*BRCA2*mut M0, n = 11) and 39 ng/mL (*BRCA2_WT_* M0, n = 80), respectively. In the M1 subgroup, mean PSA at presentation was significantly lower in the *BRCA2*mut group than the *BRCA2*_WT_ group (102 ng/mL vs. 869 ng/mL, *p* = 0.018, *t*-test with Welch’s correction, Appendix A). In the non-metastatic subgroup, contrastingly, serum PSA was not significantly different between the *BRCA2*mut group and the *BRCA2*_WT_ group (89 ng/mL vs. 39 ng/mL, *p* = 0.425, *t*-test with Welch’s correction).

### 2.4. Genomic Characteristics of BRCA2 Mutated Prostate Cancers

Common genetic alterations in prostate cancer, such *as TMPRSS2-ERG fusion, TP53, AR, MYC,* or *CDK12* were not significantly associated with *BRCA2* mutations. Enriched gene mutations (including variants of unknown significance) in *BRCA2*mut group included *ATM, APC* and *BRCA1* (Table 2, sample data available in Appendix A). Enriched gene mutations in wild-type tumors include *SPOP* (7.5% vs. 0.0%, *p* = 0.606, Fisher’s exact test) (Appendix A). Median tumor mutation burden (TMB) was calculable in 87 (69%) of the 126 sequenced samples, and microsatellite instability (MSI) was calculable in 89 (71%) of the samples. TMB of *BRCA2*mut group was significantly higher than *BRCA2*_WT_ group (12.2 vs. 3.9 mutation/Mb, *p* < 0.001, Mann-Whitney test, Appendix A). In contrast, median microsatellite instability (MSI) was not significantly different between the two groups (2.90% vs. 3.23%, *p* = 0.895, Mann-Whitney test).

### 2.5. Serum PSA Level Validation in Public Datasets

To validate our findings, we assessed four large public datasets of metastatic castration-resistant (n = 444, 123) [6,22], metastatic castration-sensitive (n = 424) [23], and localized prostate cancers (n = 187) with serum PSA data available [24]. The prevalence of BRCA2 alterations in the studies are summarized in Table 3.

#### 2.5.1. Localized BRCA2 Mutated Prostate Cancer

In the TCGA-localized prostate cancers, *BRCA2* genetic alterations (mutations or deep deletion, somatic/germline) were found in 7 (4%) patients. In this group, Serum PSA of *BRCA2*-mutated tumors was not significantly different from the wild-types (26.4 ng/mL vs. 10.4 ng/mL, *p* = 0.107, *t*-test with Welch’s correction, Figure 1), consistent with our data. Median mutation count was higher in the *BRCA*2-mutated group, yet did not reach statistical significance (30 vs. 21, *p* = 0.067, Mann-Whitney test).

#### 2.5.2. Metastatic Castration-Sensitive BRCA2 Mutated Prostate Cancer

In the mCSPC dataset (Stopsack et al.), we excluded those received androgen deprivation therapy (n = 104) for PSA level comparison. In the hormone-naïve subgroup (n = 297), Serum PSA of *BRCA2*-mutated tumors was significantly lower than the wild-types (127 ng/mL vs. 306 ng/mL, *p* = 0.031, *t*-test with Welch’s correction, Figure 1). For mutation count comparison, we used the full dataset. *BRCA2*-mutated tumors (n = 23) had higher mutation count than the wild-types (n = 401) (*p* < 0.0001, Mann-Whitney test, Figure 2), consistent with our results.

#### 2.5.3. Metastatic Castration-Resistant BRCA2 Mutated Prostate Cancer

In the first mCRPC dataset (Abida et al.), clinically significant *BRCA2* alterations were found in 47 samples (10%). Compared to those with wild-type *BRCA2* (n = 374), *BRCA2*-mutated samples had significantly lower serum PSA at presentation (48 ng/mL vs. 186 ng/mL, *p* = 0.017, *t*-test with Welch’s correction, Figure 1) and higher mutation count (*p* < 0.0001, Mann-Whitney test, Figure 2). In the second mCRPC dataset (Wei et al.), *BRCA2* alterations (somatic and germline) were found in 20 samples (19%). Similarly, their serum PSA at presentation was significantly lower than the wild-type tumors (53.4 ng/mL vs. 190 ng/mL, *p* = 0.006, *t*-test with Welch’s correction, Figure 1).

### 2.6. Genomic Characteristics Validation in Public Dataset

Using the cBioPortal, we validated the co-occurrence or mutual exclusivity of gene mutations with *BRCA2* genetic alterations in prostate cancer public datasets (6875 patients or 7151 samples from 22 studies). *BRCA2* genetic alterations were found in 383 (6%) of queried patients. Tumor mutation burden was significantly higher in the *BRCA2*-altered samples. We confirmed enriched genomic alterations of *ATM* (11% vs. 5.1%, *p* = 1.442 × 10^−5^) and *APC* (16% vs. 5.8%, *p* < 1 × 10^−10^) (Fisher’s exact test), consistent with our data. SPOP mutation was more frequent in *BRCA2*-altered group (16% vs. 9.9%, *p* = 6.079 × 10^−4^), opposite to our data.

## 3. Discussion

We found that *BRCA2*-mutated metastatic PCs present with relatively low serum PSA. In localized tumors, however, serum PSA of *BRCA2* mutated PCs was not lower than the wild-type PCs. Also, we validated our findings with multiple publicly available mCSPC and mCRPC datasets. To our knowledge, this is first report that suggests *BRCA2*-mutated metastatic tumors differ from the wild-type tumor by PSA level at diagnosis. Velho et al. reported that germline DNA-repair gene mutation-positive prostate cancer patients (52% BRCA1 or *BRCA2*) had lower median PSA levels at diagnosis than mutation-negative patients. The patient cohort analyzed by Velho et al. consisted of 53–62% of T3/4 and 14–23% M1 disease at diagnosis [25]. Based on our analysis, we speculate that the PSA-low advanced stage tumors are enriched by *BRCA2* mutation carriers. For metastatic PCs, it is now recommended by the NCCN, AUA or EAU guidelines to run both.

Serum PSA is a highly sensitive biomarker for prostate cancer diagnosis as well as monitoring for recurrence and drug response. Yet, its specificity is not high enough for metastatic disease monitoring, leaving cases with clinical or radiographic progression without PSA elevation [26]. Our findings speculate that *BRCA2*-altered prostate cancer progresses rapidly to metastatic and castration-resistant disease not necessarily accompanied by increased serum PSA level. Molecular studies suggest that *BRCA2* mutation combined with RB1 alteration, which is located closely on chromosome 13q14, can generate an aggressive PC model with castration resistance [27].

Our study shows that *BRCA2* mutation is associated with lower PSA at the metastatic state, but not in the localized setting. We suggest some explanations, but these are not yet definitive. First, the major source of serum PSA in patients with metastatic prostate cancer is the cancer cells themselves. However, in a localized setting, theoretically, PSA released from adjacent normal cells may contribute more significantly than in a metastatic setting. Secondly, genetic characteristics of *BRCA2* wild-type localized cancer may be different from those of *BRCA2* wild-type metastatic cancer. As *BRCA2* mutation is enriched in metastatic prostate cancer, *TP53* mutation or RB1 deletions are also enriched in metastatic disease, whereas *SPOP* mutation is more frequent in localized disease [6,21,22,28,29]. Lastly, the metastatic microenvironment may pressure cancer cells to express PSA more or less than the primary site. In other words, a *BRCA2*-mutated cancer cell at the primary site (prostate) may express high PSA while a cell with same genetic background but located in bone marrow may express less PSA.

Among the various DNA-repair genes of HRR or mismatch repair, we focused on *BRCA2* alteration because it is the most frequently mutated HRR gene in advanced PCs [1], and represents a subset of patients that are likely to respond to PARP inhibitors [12,13]. While earlier studies combined ATM mutation in this subgroup [4], recent evidence shows that PARP inhibitor alone is not enough to treat the ATM mutated PCs [30]. Indeed, we found that ATM mutation frequently accompanies *BRCA2*, which may explain the earlier-documented benefit of PARP inhibitors in ATM-mutated cases. *ATM* germline mutation increases the likelihood of PC development [31], and ATM mutation or protein loss are enriched in high-grade cancer [32]. In response to DNA double strand break, the ATM protein modulates diverse aspects of the cell’s responses such as cell cycle checkpoint arrest, apoptosis—not limited to DNA repair itself [33,34,35]. This may lead to varying responses to DNA-damaging agents such as PARP inhibitors or radiations in patients with ATM somatic or germline alterations unlike *BRCA1* or *BRCA2* [36,37,38].

We point to some limitations in this study. Interpretation of our data is limited due to the study’s retrospective design. Serum PSA data was available in only a subset of the datasets analyzed in this study, which may increase selection bias. We did not perform survival analysis as the consecutive treatments after sequencing analysis were rather heterogenous. Particularly, some of the patients with HRR mutations underwent clinical trials involving PARP inhibitors which would deviate survival outcome. Lastly, we tried setting some arbitrary cut-off points (100 ng/mL, 10 ng/mL) of PSA threshold or values that could possibly be considered suspicious and warrant genetic testing in metastatic PC at presentation, which were not successful. We argue that somatic and germline genetic testing is recommended in patients with metastatic PC no matter what their PSA level is.

It has been known that there is a clinically distinctive aggressive variant prostate cancer (AVPC), presenting with low-serum PSA, neuroendocrine small cell or ductal histology, which frequently metastasizes to visceral organs and rapidly develops castration resistance. Yet there is substantial genomic and transcriptomic heterogeneity that exists within these tumors [39]. Our paper addresses the contention that *BRCA2*-mutated metastatic PC fits in the criteria of AVPC, providing further understanding on this disease entity.

## 4. Materials and Methods

Patient data: The YUHS Big-Data team identified cases of prostate cancer patients who had performed next generation sequencing on their achieved prostate cancer tissues (prostatectomy, biopsy or transurethral resection specimen) between January 2019 and May 2021. Associated clinical and pathological information was retrieved from electrical medical records. The study design was approved by Severance Hospital Institutional Review Board (IRB #4-2020-0812). Patient consent was waived due to the retrospective nature of the study. All data underwent deidentification process.

Panel Sequencing of Prostate Cancer Tissues: Targeted DNA and RNA sequencing were performed using TruSight Tumor 170 or 500 (Illumina, San Diego, CA, USA). Briefly, 40 ng of formalin-fixed paraffin-embedded (FFPE) tissue-derived DNA and RNA were extracted using QIAGEN AllPrep DNA/RNA FFPE Kit (Qiagen, Hilden, Germany). After hybridization capture-based target enrichment, paired-end sequencing (2 × 150 bp) was performed using a NextSeq sequencer (Illumina) according to the manufacturer’s instructions. Variants with a total depth of at least 100× were included for analysis. Variant interpretation was based on recommendations from the Association for Molecular Pathology (AMP), American Society of Clinical Oncology (ASCO) and College of American Pathologists (CAP) [40]. AMP/ASCO/CAP Tier 1 variant included level 1 and level 2 genetic alterations that are FDA-approved biomarkers and standard of care. Tier 2 variant included alterations with compelling clinical or preclinical evidence to drug response. Variant interpretations followed the “Standards and guidelines for the interpretation of sequence variants” published by the American College of Medical Genetics and genomics (ACMG) [41]. Each variant was initially queried using Varsome (Varsome.com) which annotates variants by combining data resources including AACT Clinical Trials from clinicaltrials.gov, Pharmacogenomics Knowledge Base (PharmGKB), COSMIC by the Sanger Institute, Polyphen-2 scores by Harvard University, CADD scores by the University of Washington, Clinical Pharmacogenetics Implementation Consortium (CPIC), the Drug-Gene Interaction Database (DGIdb) and OMIM [42]. Output variant annotation was manually reviewed in OncoKB website (http://www.OncoKB.org) (last accessed on 30 September 2022).

Public Dataset Analysis: We downloaded annotated clinical and genomic data from the cBioportal—“Prostate Adenocarcinoma (TGCA, Cell 2015)”, “Metastatic castration-sensitive prostate cancer (MSK, Clin Cancer Res 2020)”, and “Metastatic Prostate Adenocarcinoma (SU2C/PCF Dream Team, PNAS 2019)”. Additionally, we downloaded annotated clinical and genomic data of metastatic castration-resistant prostate cancer from the publication of Wei et al. [6].

Statistical Methods: Cases were grouped by their germline or somatic alteration status of *BRCA2* (mutation, deletion). Serum PSA at diagnosis and tumor mutation count were compared between *BRCA2*-altered group and wild-type from each study. Baseline characteristics, including Gleason grade group, TNM stage and other gene mutations were compared between the two groups using Pearson’s chi-squared test or Fisher’s exact test. Continuous variables of age at diagnosis and PSA at diagnosis were compared using *t* test with Welch’s correction. Tumor mutation burden or total mutation count were compared using Mann-Whitney test. All *p* values were two-sided and *p* < 0.05 was defined as statistically significant. Statistical analysis was performed using SPSS 26.0 and GraphPad Prism 9.2.0.

## Figures and Tables

**Figure 1 ijms-23-13426-f001:**
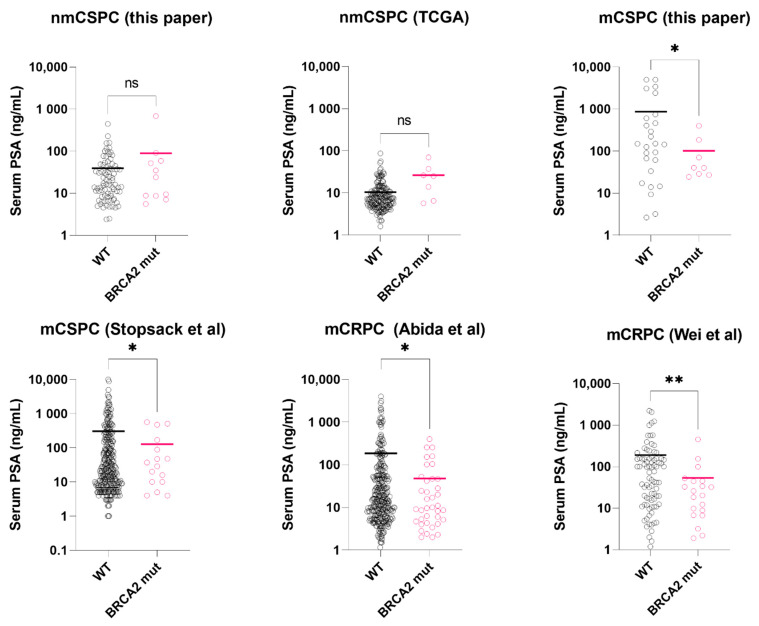
Serum PSA level at diagnosis in *BRCA2*-mutated vs. wild-type prostate cancer, stratified by metastatic stage and castration-sensitivity. Asterisks: * *p* < 0.05; ** *p* < 0.01. black circle: BRCA2 wild-type sample (WT); pink circle: BRCA2 mutated sample (mut). Colored bar = median. Datasets: This paper, TCGA. [24], Stopsack et al. [23], Abida et al. [22], Wei et al. [6].

**Figure 2 ijms-23-13426-f002:**
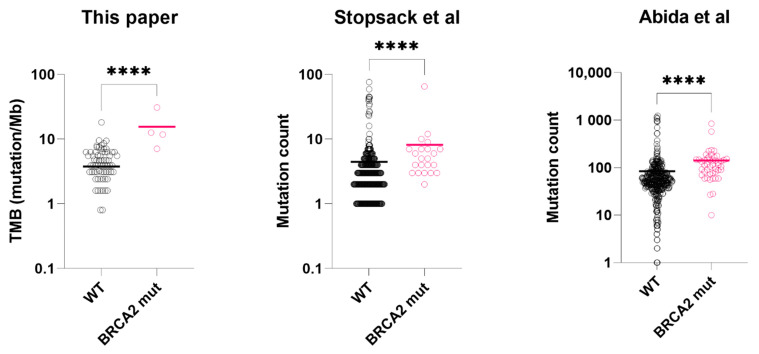
Tumor mutation burden or mutation count of *BRCA2*-mutated prostate cancer vs. wild-type prostate cancer tissues. From left to right: tumor mutation burden (mutation/Mb), data from this paper; tumor mutation count (total), data from Stopsack et al.; tumor mutation count (total), data from Abida et al. Asterisks: **** *p* < 0.0001. black circle: BRCA2 wild-type sample (WT); pink circle: BRCA2 mutated sample (mut). Colored bar = median. Datasets: This paper, TCGA. [24], Stopsack et al. [23], Abida et al. [22], Wei et al. [6].

**Table 1 ijms-23-13426-t001:** Clinical and Pathologic Characteristics of *BRCA2* mutated cancers.

	*BRCA2* Mutated N = 19	*BRCA2* Wild Type N = 107	*p* Value
Age at diagnosis (years) Mean (range)	64.1 (43–88)	66.3 (44–86)	0.432 ^1^
PSA at diagnosis (ng/mL) Mean (range)	94.7 (5.6–682)	242.7 (2.4–5000)	0.101 ^1^
cT stage, N (%)			0.925 ^2^
T2	3 (16%)	16 (15%)	
T3/4	16 (84%)	91 (85%)	
N1 stage, N (%)	6 (32%)	48 (45%)	0.324 ^3^
M1 stage, N (%)	8 (42%)	27 (25%)	0.165 ^3^
Gleason grade group, N (%) *			0.140 ^2^
1–4	6 (32%)	51 (50%)	
5	13 (68%)	51 (50%)	
Histologic Type, N (%) *			0.070 ^3^
Adenocarcinoma, acinar type	16 (84%)	86 (84%)	
Adenocarcinoma, ductal type	1 (5%)	15 (15%)	
Neuroendocrine small cell	2 (11%)	1 (1%)	
Adenosquamous	0 (0%)	1 (1%)	

^1^ unpaired *t*-test with Welch’s correction, two-tailed; ^2^ Chi-square test; ^3^ Fisher’s exact test. * Five wild type tumors were from metastatic specimen, unevaluable for GGG or histologic typing.

**Table 2 ijms-23-13426-t002:** Genomic Characteristics of *BRCA2* mutated cancers.

	*BRCA2* Mutated N = 19	*BRCA2* Wild Type N = 107	*p* Value
*TMPRSS2-ERG* fusion, N (%)	3(16%)	27(25%)	0.560 ^1^
*TP53*, N (%)	2(11%)	22(22%)	0.522 ^1^
*AR*, N (%)	7(37%)	19(18%)	0.070 ^1^
*MYC*, N (%)	7(37%)	18(17%)	0.061 ^1^
*CDK12*, N (%)	3(16%)	16(15%)	>0.999 ^1^
*FGFR1*, N (%)	1(5%)	17(16%)	0.140 ^1^
*ATM*	8(42%)	11(10%)	0.002 ^1^
*APC*	7(37%)	9(8%)	0.003 ^1^
*BRCA1*	4(21%)	4(4%)	0.018 ^1^

^1^ Fisher’s exact test.

**Table 3 ijms-23-13426-t003:** *BRCA2* mutation incidences in prostate cancer datasets with PSA level data available.

Source (Disease State)	*BRCA2* Mutated	*BRCA2* Wild Type
This paper (nmCSPC)	11 (12%)	80 (88%)
TCGA (nmCSPC)	7 (4%)	180 (96%)
This paper (mCSPC)	8 (23%)	27 (77%)
Stopsack et al. (mCSPC)	17 (5%)	303 (95%)
Abida et al. (mCRPC)	56 (15%)	321 (85%)
Wei et al. (mCRPC)	20 (19%)	83 (81%)

## Data Availability

All processed genomic, pathologic and clinical data are presented in Appendix A. Original genomic data will be provided upon request. The link and location of public data used in this paper is described in method section.

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
