# Peer review of "Characteristics of BRCA2 Mutated Prostate Cancer at Presentation"

_ijms, 2022, doi:10.3390/ijms232113426_

Round 1

Reviewer 1 Report

Dear authors, you have presented very interesting work on the clinical significance of BRCA2 mutation in advanced prostate cancer. Please consider the following suggestions

1. the introduction should not start with PARP-i as this is not the focus of your paper. My suggestion is to start the introduction with the incidence of genetic alterations in localised and metastatic CaP.

2. BRCA2 mutations in CaP are associated with better outcomes IF PARP-i are combined or followed by chemotherapy (maybe rephrase line 49)

3. You have nicely shown that BRCA2 mutation are associated with lower PSA at the metastatic state but not in the localised setting, can you explain this finding?  

4. Also is there a PSA threshold or value that could possibly be considered suspicious and warrant genetic testing in mCaP? As per NCCN guidelines all mCaP patients should undergo genetic testing, either gremlin or somatic. What is therefore the added clinical benefit of your study in terms of everyday clinical practice. Please discuss this issue in the discussion 

4. You have not included a paragraph on the limitations of your study which is mandatory especially given the nature of your research

Author Response

Dear authors, you have presented very interesting work on the clinical significance of BRCA2 mutation in advanced prostate cancer. Please consider the following suggestions

  1. the introduction should not start with PARP-i as this is not the focus of your paper. My suggestion is to start the introduction with the incidence of genetic alterations in localised and metastatic CaP.

We appreciate your positive comments on our work. Following your suggestion, we revised the introduction part and abstract to start with the incidence of genetic alterations of BRCA2 in localized and metastatic CaP.

  1. BRCA2 mutations in CaP are associated with better outcomes IF PARP-i are combined or followed by chemotherapy (maybe rephrase line 49)

You are right. We appreciate your comment and rephrased our expression in introduction part as follows: recently, polyadenosine diphosphate-ribose polymerase (PARP) inhibitors olaparib and rucaparib have demonstrated improved overall survival in BRCA2 altered metastatic PCs, combined or followed by conventional chemo-hormonal therapies.

  1. You have nicely shown that BRCA2 mutation are associated with lower PSA at the metastatic state but not in the localised setting, can you explain this finding?

There can be several explanations but anything definitive. First, the major source of serum PSA in patients with metastatic prostate cancer is cancer cell themselves. However, in localized setting, theoretically, PSA released from adjacent normal cells may contribute more significantly than in metastatic setting. Secondly, genetic characteristics of BRCA2 wild-type localized cancer may be different from those of BRCA2 wild-type metastatic cancer. As BRCA2 mutation is enriched in metastatic prostate cancer, TP53 mutation is also enriched in metastatic disease whereas SPOP mutation is more frequent in localized disease. Lastly, the metastatic microenvironment may pressure cancer cells to express PSA more or less than primary site. In other words, a BRCA2 mutated cancer cell at primary site (prostate) may express high PSA while a cell with same genetic background but located in bone marrow may express less PSA. Again, these are all possibilities. We added this part in the discussion section of the revised manuscript.

  1. Also is there a PSA threshold or value that could possibly be considered suspicious and warrant genetic testing in mCaP? As per NCCN guidelines all mCaP patients should undergo genetic testing, either gremlin or somatic. What is therefore the added clinical benefit of your study in terms of everyday clinical practice. Please discuss this issue in the discussion.

It will be a very important point if we can set a PSA threshold that warrants genetic testing in mCaP. We tried setting some arbitrary cut-off points (100ng/mL, 10ng/mL) which were not successful. We added this part in the discussion section of the revised manuscript.

  1. You have not included a paragraph on the limitations of your study which is mandatory especially given the nature of your research

We agree that a paragraph on the limitations of our study should be included. Retrospective nature is the main limitation of this study. We did not perform survival analysis as the consecutive treatments after sequencing analysis were rather heterogenous. Especially, some of the patients with HRR mutations underwent clinical trials involving PARP inhibitors which would deviate survival outcome. We added this part in the discussion section of the revised manuscript.

Reviewer 2 Report

Ms. Ref. No.: ijms-1987174

Title: Characteristics of BRCA2 mutated prostate cancer at presentation 

Multiple studies showed that BRCA2 mutations have a high risk of developing prostate cancer. This study aimed to mutational profile of BRCA2 in prostate cancer (PC) using the targeted gene panel sequencing data from 126 consecutive prostate cancer tissue. In this study, the authors found that BRCA2 mutated metastatic prostate cancer present with relatively low serum PSA. The topic described in this manuscript is interesting. 

Minor comments:

(1).    Please use a table to summarize the data from several different studies relevant to the prevalence and frequency of BRCA2 in patients with prostate cancer. I think it is more clear and helpful to the reader to understand the recent studies.

(2).    Line 232: Please describe the role ATM mutations in prostate cancer in detailed. The authors showed that ATM mutation is frequently accompanied with BRCA2 mutation (Table 1).

Author Response

(1).    Please use a table to summarize the data from several different studies relevant to the prevalence and frequency of BRCA2 in patients with prostate cancer. I think it is more clear and helpful to the reader to understand the recent studies.

We appreciate your positive comments on our work. We agree that a table summary of reported BRCA2 prevalence are necessary. We added this part in the result section (new Table 3) of the revised manuscript.

(2).    Line 232: Please describe the role ATM mutations in prostate cancer in detailed. The authors showed that ATM mutation is frequently accompanied with BRCA2 mutation (Table 1).

We agree that further description on the role of ATM mutation in prostate cancer should be included. ATM germline mutation increases likelihood of PC development, and ATM mutation or protein loss are enriched in high-grade cancer. In response to DNA double strand break, the ATM protein modulates diverse aspects of cell’s responses such as cell cycle checkpoint arrest, apoptosis – not limited to DNA repair itself. This may lead to varying responses to DNA-damaging agents such as PARP inhibitors, platinum-drugs or radiations in patients with ATM somatic or germline alterations unlike BRCA1 or BRCA2. We added this part in the discussion section of the revised manuscript.